# Holistic Token Efficient in Speculative Decoding: Post-use & Pre-cut

## Abstract

Large language models (LLMs) deliver strong generative performance but suffer from high inference latency. Speculative Decoding (SD) accelerates inference by allowing a fast draft model to propose tokens, which are then verified in parallel by a larger target model. While SD provides lossless acceleration while preserving identical generation quality, its key challenge lies in draft token efficiency: ensuring that as many drafted tokens as possible are converted into useful tokens in the final output. We present a holistic token-efficient SD strategy built on two complementary mechanisms. *Ex-post utilization (Post-use)* employs a token cache to recycle and reuse useful drafts in subsequent forward passes. *Ex-ante reduction (Pre-cut)* adaptively controls draft length, preventing overproduction when the marginal benefit falls below the cost. Together, these mechanisms both reuse what has been produced and eliminate what should not be produced. Experiments show 2.52–3.23× overall speedup over auto-regressive decoding and over 20% higher token utilization than vanilla SD methods.

## 1 Introduction

Large language models (LLMs) such as GPT-4 (OpenAI et al., 2024), Llama (Touvron et al., 2023), and DeepSeek (DeepSeek-AI et al., 2025) have greatly advanced natural language processing (Khurana et al., 2023). However, their inference remains bottlenecked by the auto-regressive nature of decoding: each token must be generated sequentially, leading to high latency. Many strategies have been explored to improve LLM efficiency, such as low-bit quantization (Zhao et al., 2024c; Huang et al., 2024b), neural pruning (Ma et al., 2023; Ling et al., 2024; Kim et al., 2024; Huang et al., 2025), and knowledge distillation (Liu et al., 2024; Yang et al., 2024). However, such approaches inherently trade off generation quality, and cannot break the auto-regressive nature of LLM inference.

Speculative Decoding (SD) (Miao et al., 2024; Xia et al., 2023; Leviathan et al., 2023b) has recently been proposed to overcome this limitation. As illustrated on the left side of Figure 1, in SD, a lightweight draft model first proposes multiple tokens, which are then verified in parallel by a larger target model. Accepted tokens have the *same quality* as if they were generated auto-regressively. Yet, the overall latency is reduced to one forward pass of the target model plus several from the draft model. This is substantially faster than using the target model alone. In this way, SD achieves *lossless acceleration*: producing identical outputs to auto-regressive decoding while significantly reducing latency.

While SD offers lossless acceleration, it has not yet fully realized its potential. In the best case, the draft model should generate exactly the tokens that the target model accepts—*no more and no fewer*. In practice, the speed up of SD (see Section 3.1.1) for a given draft/target model pair depends on three factors: (i) the latency ratio of draft vs. target model for a single forward call, (ii) the draft utilization (i.e., proportion of draft tokens appeared in final output), and (iii) the mean accept length (MAT, i.e., the average number of draft tokens accepted at each verification step). Most prior work focuses on building faster draft models. In contrast, we ask: *how can we make every drafted token useful and reduce the number of verification calls?* Put simply, our goal is to maximize both draft utilization and MAT at the same time. To achieve this goal, we pursue two complementary mechanisms: *Ex-post utilization (Post-use)*, which increases the yield from already generated drafts

Figure 1: Comparison between vanilla auto-regressive decoding (upper left), vanilla Speculative Decoding (lower left), and our proposed framework (right).

via token caching and reuse; and *Ex-ante reduction (Pre-cut)*, which curbs drafting at the source by adaptively stopping when the marginal benefit no longer justifies the cost.

We develop an on-the-fly SD plug-in that optimizes both directions and can be integrated into existing transformer-based SD paradigms.

For **Ex-post utilization (Post-use)**, we first redefine the USEFUL tokens after verification (see Equation 7), as we found that such USEFUL tokens may be re-generated in the subsequent drafting process especially when they are not used in final output (see Section 4.1), like the "recycled" tokens marked in Figure 1. To exploit this redundancy, we introduce a token cache. The cache stores USEFUL tokens in an indexed pool (short $n$-grams keyed by their first token) and reuses them during subsequent draft/target passes. During a forward call, cached tokens are fetched and checked. If a match is found, the matched tokens are directly appended to the sequence, thereby generating extra tokens in a single forward pass. If a match is not found, decoding proceeds as usual. The cache is managed with lightweight ranking, eviction, and refresh to keep it small and efficient.

While Post-use increases the value of tokens already generated, **Ex-ante reduction (Pre-cut)** reduces what should not be generated. We employ a reinforcement-learning based controller that monitors the drafting process online and issues CONTINUE/STOP decisions, halting drafting when the predicted marginal gain in useful (or recyclable) tokens falls below the marginal cost. The decision is made by a $\mathcal{Q}$-Table that learns the relationship between simple signals—recent acceptance statistics, cache hit rate, and observed cost slope—and benefits gained after verification.

Together, Post-use reuses what is produced and Pre-cut cuts what should not be produced, yielding a holistic improvement in token efficiency. In experiments across diverse model pairs and tasks, the framework achieves 2.52–3.23× overall speedup over auto-regressive decoding and delivers token utilization gains of over 20% compared to prior SD methods. To summarize, our contributions are as follows.

- We redefine USEFUL tokens after verification and build a token cache that stores such USEFUL tokens. We introduce lightweight ranking, admission, and eviction policies to maintain a compact cache and prioritize look-ups for high-reuse candidates.

- We design a token cache reuse mechanism that reduces the number of draft-model calls by avoiding the regeneration of tokens that are likely to recur, thereby improving the effective utilization of draft tokens. Before each draft/target forward, we fetch the cache candidates with the current anchor token (i.e., the list token of the prefix) and, on a hit, append the matched tokens into the sequence for lossless verification.

- We establish a connection between online signals from the drafting process (recent acceptance statistics, cache-hit rate, and cost slope) with a reinforcement-learning controller. The controller utilizes a $\mathcal{Q}$-Table to predict the benefits that can be gained from the current draft, and stops drafting when the predicted marginal benefit falls below the marginal cost.

The rest of this paper is organized as follows: Section 2 reviews related work on LLM efficiency and Speculative Decoding. Section 3 introduces the background of Speculative Decoding and Q-learning. Section 4 introduces the motivation and observation on which our proposed solution is based. Section 5 presents the framework we proposed in detail. Section 6 describes the experimental setup and results. Finally, Section 7 concludes the paper.

## 2 RELATED WORK

### 2.1 TRAINING DRAFT MODELS

One key research area in SD is to train an efficient draft model. The draft model can be an independent model with significantly fewer parameters than the target model. It can be obtained by: (i) *training from scratch* (e.g., SpecDec (Huang et al., 2024a)) or (ii) *distillation* from the target model (e.g., DistillSpec (Zhou et al., 2023)). Moreover, semi-independent draft models can retrieve the target model's prediction for multiple future tokens from its intermediate representations. For instance, Eagle-1 (Li et al., 2024b) and Eagle-3 (Li et al., 2025) utilize an auto-regressive layer to retrieve hidden state information. In contrast, Glide (Du et al., 2024) retrieves from the target model's key-value cache, and Medusa (Cai et al., 2024) uses multiple prediction heads to generate draft tokens.

### 2.2 OPTIMIZING THE SD PARADIGM

Another orthogonal research area in SD is to optimize the cooperation between the draft and target models; this is where our work lies. Typically, a smaller model from the same family as the target model is used as the drafter. Such a drafter can produce high-quality drafts but is not as fast as specialized drafters, like Eagle (Li et al., 2025); hence, careful control of drafting is required to maximize token efficiency. Many studies have explored adaptive drafting (Fu et al., 2024a; Brown et al., 2024) Some works directly leverage the draft/target model's output information (Li et al., 2024a; Gante, 2023) while others introduce an extra module to evaluate the optimized draft length, such as a perceptron (Zhang et al., 2024), a ResNet (Huang et al., 2024a), or trainable prediction heads (Du et al., 2024). PEARL (Liu et al., 2025) utilizes additional GPUs to run the draft and target models in parallel, enabling asynchronous drafting and verification. Meanwhile, works such as Lookahead (Fu et al., 2024b; Zhao et al., 2024b), and SkipDecode (Corro et al., 2023), have shown that token segments from prior inference runs can be reused to accelerate LLM inference and achieved a satisfying speedup with low overhead. More recently, Ouroboros (Zhao et al., 2024a) has brought their idea of token reuse into Speculative Decoding to generate longer drafts.

Most prior efforts focus on per-iteration optimizations of SD (e.g., threshold control or local token reuse). In contrast, we take a global view and manage token efficiency across iterations via Post-use and Pre-cut. This training-free, drop-in perspective complements existing SD frameworks and appears underexplored.

## 3 BACKGROUND

This section introduces the background of this paper. Starting with an introduction of the Speculative Decoding (SD) paradigm as well as our optimizing object, and then the definition of Q-learning that the pre-cut is based on. The notations used in this paper are listed and explained in Appendix B.

### 3.1 SPECULATIVE DECODING

Speculative Decoding (SD) follows a *draft–verify* paradigm to accelerate inference while preserving generation quality (Leviathan et al., 2023a). Here, we detail the two key stages of SD: Drafting and Verifying.

**Drafting:** Given the input sequence context $[x_1, \ldots, x_n]$, the draft model $\mathcal{M}_d$ predicts the probability distributions for the next $\lambda$ tokens:

$$\widetilde{p}_{n+1}, \ldots, \widetilde{p}_{n+\lambda} = \mathcal{M}_d\big([x_1, \ldots, x_n], \lambda\big), \tag{1}$$

from which draft tokens $\widetilde{d}_{n+1}, \ldots, \widetilde{d}_{n+\lambda}$ are greedily sampled.

**Verification:** The target model $\mathcal{M}_t$ then evaluates the extended sequence with draft tokens:

$$p_{n+1}, \ldots, p_{n+\lambda} = \mathcal{M}_t\big([x_1, \ldots, x_n, \widetilde{d}_{n+1}, \ldots, \widetilde{d}_{n+\lambda}]\big). \tag{2}$$

For each position $i$, the draft token $\widetilde{d}_i$ is accepted if it match with the greedily sampled token $d_i$ from target model's distribution that $d_i = \widetilde{d}_i$, Otherwise, $\mathcal{M}_t$ replaces the first mismatched token and resumes decoding. Importantly, as the acceptance criteria ensure the final generated sequence is identical to decoding directly with $\mathcal{M}_t$ alone, SD ensures *lossless acceleration*.

### 3.1.1 OBJECTIVE FORMULATION

The theoretical speed-up ratio $S$ of the Speculative Decoding framework can be computed with several key information:

$$\frac{1}{S} = \frac{\frac{L}{\text{MAT}} \times t_{\text{Target}} + \frac{L}{\text{Util}} \times t_{\text{Draft}}}{L \times t_{\text{Target}}} = \frac{1}{\text{MAT}} + \frac{1}{\text{Util}} \cdot \frac{t_{\text{Draft}}}{t_{\text{Target}}} \tag{3}$$

Here, $L$ is the total number of tokens to be generated, MAT is the mean acceptance length (i.e., number of drafts accepted after each verification), Util is the draft utilization rate, which is $L$ / # of $\mathcal{M}_d$ calls, $t_{\text{Draft}}$ and $t_{\text{Target}}$ are the single token latency of draft model and target model respectively. Hence, our optimization target is to maximize MAT (no fewer) and draft utilization Util (no more).

## 3.2 Q-LEARNING

Reinforcement Learning (RL) is a machine learning method where an agent learns by interacting with its environment. At each step, the agent observes the current state $s_t$, chooses an action $a_t$, and receives a reward $r_t$. The goal is to find the best strategy (i.e., policy $\pi^*$) that maximizes total future rewards, calculated as:

$$G_t = \sum_{k=0}^{\infty} \gamma^k r_{t+k} \tag{4}$$

Here, $G_t$ represents the cumulative reward, $\gamma$ is the discount factor (ranging from 0 to 1) that reduces the importance of distant rewards, and $r_{t+k}$ are future rewards.

Q-learning is a model-free reinforcement learning algorithm that evaluates the long-term value of state-action pairs $(s_t, a_t)$ by constructing a Q-table $\mathcal{Q}$. The algorithm adopts an $\epsilon$-greedy policy: with probability $\epsilon$, it randomly explores new actions; with probability $(1 - \epsilon)$, it chooses the optimal action $\arg\max_{a_t} \mathcal{Q}(s_t, a_t)$ based on the current Q-table. After executing action $a_t$, the algorithm receives immediate reward $r_t$ and observes new state $s'$, then updates the Q-value through the Bellman equation (Dietterich, 2000):

$$Q(s, a) \leftarrow Q(s, a) + \alpha \left[r + \gamma \max_{a'} Q(s', a') - Q(s, a)\right] \tag{5}$$

where $\alpha$ is the learning rate and $\gamma$ is the discount factor. This iterative update mechanism enables the Q-table to converge to the optimal policy, achieving long-term reward maximization.

## 4 MOTIVATION AND OBSERVATION

This section introduces the motivation behind our research and the observations on which our proposed solution is based. Our work is motivated by the substantial token wastage existing in current SD solutions, as the optimized draft length varies per SD iteration (i.e., drafting process and one verification behind), and many abandoned draft tokens may be regenerated. In the meantime, we refer to the observation from previous work regarding the connection between draft tokens' confidence scores and acceptance probabilities, and expand it to a metric for estimating the value of the entire draft sequence.

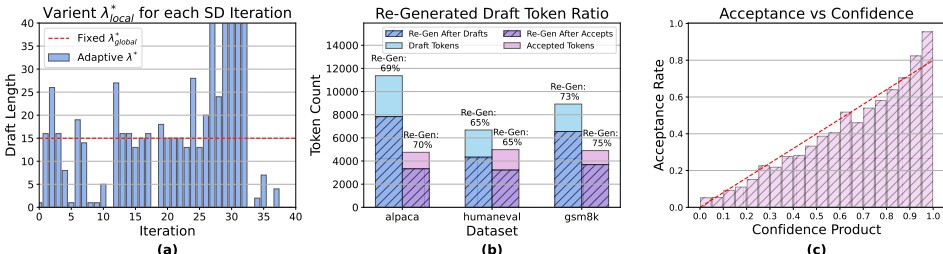

Figure 2: (a) The local optimal draft length $\lambda^*$ exhibits significant variation across different SD iterations. (b) A substantial portion of draft tokens from prior iterations can be legitimately reused. (c) Correlation between the product of the draft sequence's confidence score and the accept rate of the draft sequence

### 4.1 OPTIMAL DRAFT LENGTH AND TOKEN REUSE POTENTIAL

SD typically employs a fixed draft length $\lambda$, where the draft model generates $\lambda$ tokens for verification during each SD iteration. However, empirical observations demonstrate that optimal draft length $\lambda^*_{\text{local}}$ locally frequently differs substantially from the optimal draft length $\lambda^*_{\text{global}}$ globally (see Figure 2a). This difference leads to either redundant draft model operations (i.e., $\lambda^*_{\text{global}} \ll \lambda^*_{\text{local}}$, draft tokens are wasted) or redundant target model computations (i.e., $\lambda^*_{\text{global}} \gg \lambda^*_{\text{local}}$, more verifying operation required). Meanwhile, we have also observed that a large portion of draft tokens (in the form of phases longer than 2) from prior iterations can be legitimately reused, as illustrated in Figure 2b, a considerable proportion of draft token sequences from proceeding iterations reappear after each draft model call and target model call (i.e., after all accepted tokens). This indicates that the draft model can recycle and reuse these tokens to reduce the cost of the draft.

### 4.2 ACCEPTANCE RATES OF DRAFT SEQUENCE

To apply Ex-ante reduction (Pre-cut), we need to estimate the expected acceptance probability at a low cost. We have conducted experiments on the *alpaca* dataset to explore the relationship between the product of the draft tokens' confidence score $\prod_{i=0}^{\lambda} \text{conf}(d_i)$ (i.e., the product of output probability of LLM w.r.t. each token) and the acceptance probability $P_{\text{Accept}}(d_{0\cdots\lambda})$. As shown in Figure 2c, there is a strong positive correlation between the product of the draft tokens' confidence score $\prod_{i=0}^{\lambda} \text{conf}(d_i)$ and the acceptance probability $P_{\text{Accept}}(d_{0\cdots\lambda})$ of the draft sequence, which can be expressed as:

$$P_{\text{Accept}}(d_{0\cdots\lambda}) \sim \prod_{i=0}^{\lambda} \text{conf}(d_i) \tag{6}$$

Therefore, we can use the draft sequence's confidence score to estimate expected acceptance probability without additional overhead (See Section 5.2). Similar phenomena are observed in other methods, such as GLIDE (Du et al., 2024) and Eagle-2 (Li et al., 2024a), but they typically focus on single draft tokens.

## 5 METHOD

This section presents our proposed method, where we formulate Speculative Decoding as a resource management problem. Generating tokens incurs a computation cost due to the draft/target model's forward call. At the same time, its utility is revealed only after target-model verification. We design Ex-post Token Utilization and Ex-ante Token Reduction strategies to optimize the trade-off between draft cost and verification benefits, thereby enhancing overall token efficiency.

## 5.1 Ex-post Token Utilization

Section 4.1 demonstrates that a considerable amount of prior tokens reappear in the following generation process, hence indicating high reuse potential. The token recycling process requires identifying, collecting, and efficiently managing useful tokens. The token cache is first collected from recent draft tokens, especially for those passed target model's verification (as shown in Equation 7, and *recycled* tokens marked in Figure 1), recycled tokens are stored in a $n$-gram token cache $\mathcal{K}$ consists of a series of phases stacks (i.e., last in, first out) like $(\tilde{c}_0|\tilde{c}^1_{1,\cdots,\beta}, \tilde{c}^2_{1,\cdots,\beta})$ indexed by first key token $c_0$, and the maximin length of each phase is $\beta$ (e.g. 5).

$$\textsc{Useful}(\tilde{d}_{1,\dots,\lambda}, d_{1,\dots,\lambda}) = (d_i|\tilde{d}_i = d_i) \tag{7}$$

The token cache is implemented with a series of stacks and managed by CPU, hence to guarantee the efficiency of token cache indexing we set limit of token cache pool size and phase stack size for each index to $\kappa_{\text{pool}}$ and $\kappa_{\text{phase}}$ respectively (e.g. $\kappa_{\text{pool}} = 1000$ and $\kappa_{\text{phase}} = 20$) considering the hardware resources, once the limitation is reached, we evict the least used phase index in the cache pool, and the earliest inserted phase in the stack.

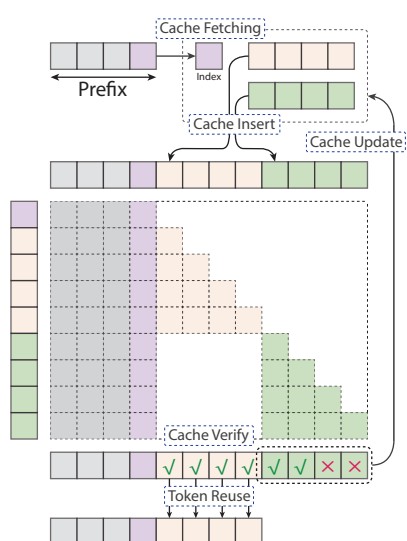

At the forward pass of the draft/target model, token caches can be natively checked for reuse by the draft/-target model. Taken draft model as an example, before the forward pass start, the module retrieves first $W$ candidate phases $(\tilde{c}_0|\tilde{c}^1_{1\dots\beta}\cdots\tilde{c}^W_{1\dots\beta})$ of length $\beta$ from the token cache whose index $\tilde{c}_0$ matches the last tokens of the prefix ($\tilde{d}_t$ for draft model and $\tilde{d}_\lambda$ for target model). Then, we appended retrieved token caches behind and form the token sequence and go through the forward pass of draft model as shown in Figure 3, a specific designed attention mask allows draft model to analysis reuse potential of each token cache independently, after the forward pass, just like the verification process we mentioned In Section 3.1, as shown in Equation 8, we can get verified result as well as a newly generated draft token $\tilde{d}_{t+1}$.

Figure 3: Token reuse & update in Ex-post Token Utilization

$$\tilde{d}_{0\dots t}, \tilde{c}^1_{1\dots\beta}\cdots\tilde{c}^W_{1\dots\beta} \xrightarrow{\text{Draft Model}} \tilde{d}_{0\dots t}, c^1_{1\dots\beta}\cdots c^W_{1\dots\beta}, \tilde{d}_{t+1} \tag{8}$$

Now, for verified token caches $c^1_{1\dots\beta}\cdots c^W_{1\dots\beta}$ we define reuse length $A_i$ as the number of matched token between $c^i_{1\dots\beta}$ and $\tilde{c}^i_{1\dots\beta}$ and the maximin reuse length $\hat{A} = \max A_i$, if we get $\hat{A} = A_i > 1$, we can append $c^i_{1\dots\hat{A}}$ to the current draft, hence we obtain two draft tokens in one draft model forward pass, other wise if $A_i \leq 1$, we append $\tilde{d}_{t+1}$ as what we usually do for SD.

For the cache reuse module attached to the target model, we select last $K$ tokens (e.g. $K = 3$) and attached 3 groups of indexed token cache to them respectively, these process enlarge the probability of token reuse in this stage, since the final output tokens should be continuous accepted tokens, hence after target model's forward pass, if the last accepted token are within those $K$ tokens, we can further append hit token cache after the accepted draft tokens, hence we got additional generated tokens during the verification process.

## 5.2 Ex-ante Token Reduction

In addition to ex-post token utilization, we introduce an ex-ante reduction strategy to prevent unnecessary drafts before verification. This goal is achieved by establishing a fine-grained mapping from draft status — including current draft cost and expected benefits — to verification results, namely useful tokens as defined in Equation 7. To formalize this, we define the token efficiency $\eta(\tilde{d}_{0\dots\lambda})$ — as the number of useful tokens produced per unit cost:

$$\eta(\tilde{d}_{0\dots\lambda}) = \frac{\text{size of } (\text{useful}(\tilde{d}_{0\dots\lambda}))}{\text{cost}(\tilde{d}_{0\dots\lambda})}$$

Here, *useful* counts tokens that satisfy Equation 7, while *cost* measures the draft model calls plus one target model call (normalized to several draft model calls). At runtime, the RL-based controller decides whether to **continue** or **stop** drafting, adjusting the draft length ($\lambda$) according to cost (number of $\mathcal{M}_d$ calls used) and expected benefit (use the confidence product in Section 4.2). Formally, the controller maintains a $\mathcal{Q}$-table $Q((\text{cost}, \text{expected}), a)$, with actions $a \in \{\text{CONTINUE}, \text{STOP}\}$. The decision rule is:

$$a = \arg \max_a Q((\text{cost}, \text{expected}), a).$$

After verification, the controller receives a reward signal for updating the control strategy:

$$r_t = \eta(\tilde{d}_{0...\lambda}) - 1$$

Subtracting 1 normalizes the baseline: if the efficiency is exactly one useful token per unit cost, the reward is zero; only when useful tokens outweigh cost is positive feedback given, while inefficient drafts yield negative feedback.

This adaptive control ensures that drafting is terminated once the marginal gain in recyclable tokens no longer outweighs the marginal cost. In this way, ex-ante reduction complements ex-post utilization, achieving a more holistic token efficiency.

## 6 EVALUATION

In this section, we first examine the overall performance of our methods in comparison to four *training-free and lossless* baselines for *general* LLM inference acceleration. Then, we conduct a detailed ablation study to investigate how our method improves Speculative Decoding via both Ex-post token reuse and Ex-ante draft reduction.

### 6.1 EXPERIMENT SETTING

**Hardware:** All the experiments are conducted on a machine with $4 \times$ A6000 GPUs, each with 48GB memory. **Model Pairs:** We select widely used LLM families to investigate the effectiveness of our proposed method, including CodeLlama (Roziere et al., 2023), Deepseek-Coder (Guo et al., 2024), Llama 2 (Touvron et al., 2023), and Llama 3.1 (Grattafiori et al., 2024). Models with a size less than 7B are used as draft models, and those with a size greater than 33B are used as target models. **Datasets:** We use various datasets that represent different types of inference tasks, including: HumanEval (code generation task) (Chen et al., 2021), MT-bench (multi-round dialogue task) (Zheng et al., 2023), GSM8K (math reasoning task) (Cobbe et al., 2021). **Baselines:** We implement four training-free lossless inference acceleration methods as our baselines, including (i) Lookahead Decoding (Lookahead) (Fu et al., 2024b) which cache recent tokens to reduce LLM inference cost, (ii) Speculative Decoding (Speculative) (Chen et al., 2023) vanilla version of SD, which generates a fixed number of draft tokens in each iteration, (iii) Assisted Generation (Assisted) (Gante, 2023) SD implementation in HuggingFace transformer, which employs naive adaptive draft length, (iv) Ouroboros (Zhao et al., 2024a), which proposes a phrase candidate pool from the verification process to generate longer drafts. All the baselines are executed with their default or universal hyper-parameters, details are listed in Appendix D. For fairness, we use greedy decoding for all the experiments to ensure all the SD approaches are lossless. And there are also some orthogonal methods like Eagle-3 (Li et al., 2025) and PEARL (Liu et al., 2025), which can be combined with our method, we also studied the potential combination performance, details are in Appendix **??**. More information about the experiment setting is described in Appendix D.

### 6.2 OVERALL PERFORMANCE STUDIES

We studied the overall performance of our method compared with the baselines on different model-pairs (using *humaneval* dataset) and different tasks (using *mt-bench* and *GSM8K* dataset), the results are shown in table 1 and table 2.

In this set of experiments, we evaluate our method against established baselines in various model pairs and tasks. We first assess model compatibility across diverse LLM pairs on the *Humaneval* dataset. As shown in Table 1, our method consistently outperforms most of the baseline methods across various model pairs. To further investigate the capability of our method, we examine task

| Methods | CodeLlama 7&34B | | CodeLlama 7&70B | | Llama2 7&70B | | Llama3.1 8&70B | |
|---|---|---|---|---|---|---|---|---|
| | tok/s | ratio | tok/s | ratio | tok/s | ratio | tok/s | ratio |
| Auto-regressive | 8.98 | 1.00× | 4.56 | 1.00× | 4.57 | 1.00× | 4.54 | 1.00× |
| Lookahead | 9.86 | 1.10× | 6.81 | 1.49× | 7.31 | 1.60× | —[1] | —[1] |
| Assisted | 16.94 | 1.89× | 11.15 | 2.45× | 11.25 | 2.46× | 12.48 | 2.75× |
| Speculative | 17.96 | 2.02× | 10.36 | 2.27× | 10.56 | 2.31× | 11.73 | 2.58× |
| Ouroboros | 22.32 | 2.49× | 12.85 | 2.82× | 13.93 | 3.05× | —[1] | —[1] |
| **Our method** | **22.64** | **2.52×** | **13.00** | **2.85×** | **14.78** | **3.23×** | **15.36** | **2.82×** |

Table 1: Comparison of baseline and our method across different model pairs on the coding task. Each cell shows throughput (tok/s) and speedup ratio.

| Methods | MT Bench | | | | GSM8K | | | |
|---|---|---|---|---|---|---|---|---|
| | Llama2 7&70B | | Llama3.1 8&70B | | Llama2 7&70B | | Llama3.1 8&70B | |
| | tok/s | ratio | tok/s | ratio | tok/s | ratio | tok/s | ratio |
| Auto-regressive | 4.54 | 1.00× | 4.16 | 1.00× | 4.18 | 1.00× | 4.12 | 1.00× |
| Lookahead | 6.25 | 1.37× | —[1] | —[1] | 6.06 | 1.45× | —[1] | —[1] |
| Assisted | 10.34 | 2.27× | 11.12 | 2.67× | 9.96 | 2.38× | 10.10 | 2.45× |
| Speculative | 11.57 | 2.54× | 10.45 | 2.51× | 10.47 | 2.50× | 10.69 | 2.59× |
| Ouroboros | 10.52 | 2.31× | —[1] | —[1] | 10.89 | 2.72× | —[1] | —[1] |
| **Our method** | **11.97** | **2.64×** | **12.43** | **2.98×** | **11.99** | **2.87×** | **12.35** | **2.99×** |

Table 2: Comparison of baseline and our method across different model-pairs on multi-round dialogue (MT Bench) and math reasoning (GSM8K) tasks. Each cell shows throughput (tok/s) and speedup ratio.

generalization with Llama 2 Chat (7B/70B) models across multi-round dialogue and math reasoning tasks, as shown in Table 2. Our method demonstrates stable performance improvement ratios across various tasks and categories, thanks to the Holistic Token Efficiency mechanism. Collectively, these experiments confirm the robustness of our method in terms of cross-model and cross-task generalization.

## 6.3 ABLATION STUDY

In this set of experiments, we conduct ablation studies to investigate how our method improves token efficiency in Speculative Decoding via Ex-post token reuse and Ex-ante draft reduction step by step.

| Levels | Speed Up Ratio | | | | | |
|---|---|---|---|---|---|---|
| | $\lambda=3$ | $\lambda=6$ | $\lambda=9$ | $\lambda=12$ | $\lambda=18$ | $\lambda=24$ |
| Auto regressive | 1.00 × | 1.00 × | 1.00 × | 1.00 × | 1.00 × | 1.00 × |
| + Speculative | 2.29 × | 2.67 × | 2.68 × | 2.55 × | 2.25 × | 2.04 × |
| + Ex-post reuse at $\mathcal{M}_d$ | 2.33 × | 2.93 × | 2.99 × | 3.19 × | 3.09 × | 2.63 × |
| + Ex-post reuse at $\mathcal{M}_t$ | 2.85 × | 3.19 × | 3.28 × | 3.34 × | 3.16 × | 2.73 × |
| + Ex-ante Reduction | 2.87 × | 3.25 × | 3.32 × | 3.42 × | 3.37 × | 3.40 × |

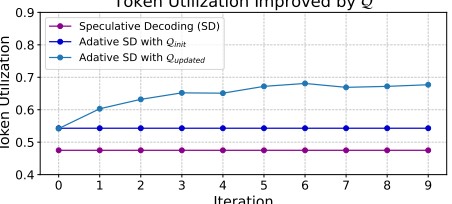

Table 3: Ablation Study

Figure 4: Ex-ante Reduction on vanilla SD

We first focus on the Ex-post token reuse module. We conduct ablation studies with Llama 2-Chat 7/70B on the *humaneval* datasets to investigate the contribution of each stage to the overall performance. Table 3 shows the contribution of different parts. Starting from the vanilla Speculative Decoding with various fixed draft length $\lambda$, we can see the optimal $\lambda$ lies between 6 to 9, as all drafts need to be generated one by one, when we add the Ex-post token reuse at draft model $\mathcal{M}_d$ and target model $\mathcal{M}_t$, the speed up ratio can be further improved and the optimal draft length $\lambda$

---

[1]Lookahead and Ouroboros are developed over transformer 4.34, while Llama 3.1 requires transformer 4.43 or above.

shifts to a larger value, as the Ex-post token reuse can efficiently reduce the latency of generating longer draft, hence leading to longer accepted length. In the meantime, with the increase of $\lambda$, the speedup gain mainly comes from the module attached to $\mathcal{M}_d$, as the number of $\mathcal{M}_d$ calls gradually becomes much larger than $\mathcal{M}_t$ calls. Finally, we add an Ex-ante draft reduction module to control over drafting, thereby stabilizing performance by reducing the overhead of useless draft tokens. The ablation study results show that each module in our method can effectively improve token efficiency in Speculative Decoding, thereby enhancing the overall inference speed.

## 6.4 EFFECTIVENESS OF EX-ANTE DRAFT REDUCTION

Now we dive into the Ex-ante draft reduction module, the control strategy, implemented with $\mathcal{Q}$-learning. The $\mathcal{Q}$-Table will be initialized with prior knowledge and updated during the inference process, Figure 4 shows the evolution of token utilization rate on SD, three lines represent vanilla SD, adaptive SD based on initialized $\mathcal{Q}$ ($\mathcal{Q}_{\text{Init}}$), and $\mathcal{Q}$ with update ($\mathcal{Q}_{\text{updated}}$) on same questions in dataset *alpaca*. The experiment results show that an initialized $\mathcal{Q}$ can achieve a higher token utilization rate than the vanilla SD, and with an adaptively updated $\mathcal{Q}$, this rate can be further improved. This indicates that the Ex-ante draft length reduction module itself can effectively enhance the token utilization rate by instantly adapting draft length to the ongoing task. We also study the variance of optimal draft length $\lambda^*$ across different model pairs and tasks with only the Ex-post token reuse module enabled (i.e., without Ex-ante draft length reduction) on Llama-2 7/70B-Instruct with the *humaneval* and *mt-bench* datasets. As shown in Figure 5, two plots demonstrate that the optimal draft length $\lambda^*$ for different model-pairs/tasks varies. This experiment emphasizes the necessity of an Ex-ante draft length reduction module for adaptive drafting.

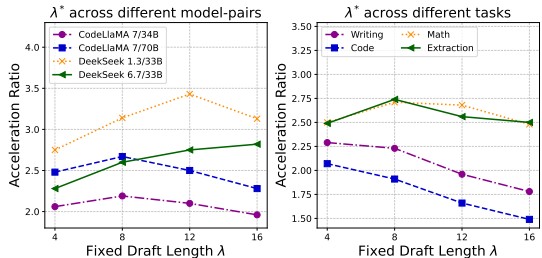

| Model Pair | Speculative | Ours |
|---|---|---|
| CodeLlama 7&34B | 8.18/68.2% | 8.27/72.3% |
| CodeLlama 7&70B | 5.44/45.4% | 7.12/62.3% |
| Llama 2 7&70B | 5.72/47.5% | 7.53/67.7% |
| Llama 3.1 8&70B | 6.83/56.9% | 7.94/64.2% |
| DeepSeek 1.3&33B | 7.09/59.1% | 7.56/66.8% |
| DeepSeek 6.7&33B | 8.02/66.9% | 8.37/73.9% |

Figure 5: Draft Length ($\lambda$) Sensitivity with Ex-post Token Reuse

Table 4: Comparison of token mean acceptance length/utilization rate

## 6.5 MEAN ACCEPTED TOKEN LENGTH AND TOKEN UTILIZATION

Finally, we study the mean acceptance length and token utilization rate of our method compared with vanilla SD method on dataset *alpaca* when both Ex-post token reuse and Ex-ante draft length reduction modules are enabled. The results are shown in Table 4. We can see that our method achieves a higher token utilization (i.e., fewer $\mathcal{M}_d$ calls) and compatible mean accept length (i.e., fewer $\mathcal{M}_t$ calls) compared with vanilla SD, which means our main objective mentioned in Section 3.1.1 is achieved. The experiment results also show that closer draft/target model sizes and newer model pairs will have a higher token utilization rate and mean accept length.

## 7 CONCLUSION

In this paper, we introduce our holistic token efficiency management strategy into Speculative Decoding. We combine Ex-post token reuse and Ex-ante draft length reduction to manage the generated useful drafts and prevent the creation of useless drafts. The experimental results demonstrate the broad compatibility of our method across different model pairs and tasks, achieving 2.52-3.23× speedup on LLM inference compared with auto-regressive decoding and higher token efficiency compared with the prior SD approach. We conducted detailed ablation studies to investigate how these two modules take effect inside the SD paradigm, and our main objective is achieved as both token utilization and mean accept length are improved with our approach. This token efficiency

analysis and management strategy can be widely integrated into existing draft models and the SD paradigm to make them exhibit their full potential.

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

# APPENDIX

## A  LLM USAGE DESCRIPTION

In the writing of this paper, Large Language Models are used in following ways:

- Correct grammar and spelling errors.
- Improve the plots used in the paper by changing colors, fonts and size.
- Format complex tables and equations to make them tidy.

## B  NOTATION

| Part | Notation | Explanation |
|------|----------|-------------|
| Speculative Decoding | $\mathcal{M}_d$ | Draft model |
| | $\mathcal{M}_t$ | Target model |
| | $\lambda$ | Draft length (number of draft tokens per iteration) |
| | $\tilde{d}_i$ | $i$-th draft token |
| | $d_i$ | $i$-th verified/accepted token |
| | $\tilde{p}_i$ | Draft model output probability for $i$-th token |
| | $p_i$ | Target model output probability for $i$-th token |
| Ex-Post Reuse | $\mathcal{K}$ | Token cache pool |
| | $\beta$ | Cached phase length |
| | $K$ | Maximum cached phase number per anchor |
| | $\tilde{c}^j_{1\cdots\beta}$ | $j$-th cached draft token sequence |
| | $c^j_{1\cdots\beta}$ | $j$-th verified cached token sequence |
| | $\kappa_{\text{pool}}/\kappa_{\text{phase}}$ | Max pool/phase size |
| Ex-Ante Reduce (RL) | $s_t$ | State at step $t$ |
| | $a_t$ | Action at step $t$ (CONTINUE/STOP) |
| | $r_t$ | Reward at step $t$ |
| | $\mathcal{Q}$ | Q-table |
| | $\gamma$ | Discount factor (RL) |
| | $\alpha$ | Learning rate (RL) |
| | $\eta(\cdot)$ | Token efficiency ratio |
| | $\text{conf}(\tilde{d}_{1\cdots\lambda})$ | Confidence score of given draft |
| | MAT | Mean accepted token length per iteration |
| | Util | Token utilization rate (accepted/generated) |

Table 5: Notations used in this paper

## C  DETAILED ALGORITHM OF OUR METHOD

The our methodframework integrates ex-ante reduction and ex-post utilization to optimize speculative decoding. Algorithm 1 outlines the complete process, which dynamically adjusts draft length via reinforcement learning and maximizes token reuse through a two-level cache system.

---

**Algorithm 1** our method: Speculative Decoding with Ex-ante Reduction and Ex-post Utilization

---

1: **function** OUR METHOD($\mathcal{M}_d, \mathcal{M}_t, \textbf{prefix}, \mathcal{Q}, \mathcal{K}$)
2:                      $\triangleright$ $\mathcal{M}_d$: draft model, $\mathcal{M}_t$: target model, $\mathcal{Q}$: Q-table, $\mathcal{K}$: token cache
3:      **while** len(**prefix**) < max_len **do**
4:          **draft** $\leftarrow \emptyset$
5:                      $\triangleright$ **Phase 1: Ex-ante Draft Reduction (RL Controller)**
6:          **for** $i = 1$ to $\lambda_{\max}$ **do**
7:                          $\triangleright$ Ex-post reuse at draft model
8:             **if** ExPostReuse(**prefix** + **draft**, $\mathcal{M}_d, \mathcal{K}$) succeeds **then**
9:                 Append reused tokens to **draft** and continue
10:             **end if**
11:             Generate next draft token $\tilde{d}_i$ from $\mathcal{M}_d(\textbf{prefix} + \textbf{draft})$
12:             **draft** $\leftarrow$ **draft** $+ [\tilde{d}_i]$
13:                               $\triangleright$ RL decision making
14:             $s_i \leftarrow$ GetCurrentState($i, \text{conf}(\tilde{d}_{1\ldots i}), \ldots$)
15:             $a_i \leftarrow \arg\max_a \mathcal{Q}(s_i, a)$
16:             **if** $a_i = a_{\text{stop}}$ **then**
17:                 **break**                     $\triangleright$ Terminate drafting based on policy
18:             **end if**
19:          **end for**
20:                       $\triangleright$ **Phase 2: Verification & Ex-post Utilization**
21:                          $\triangleright$ Ex-post reuse at target model
22:          (**verified_draft**, **reused_suffix**) $\leftarrow$ ExPostReuse(**prefix** + **draft**, $\mathcal{M}_t, \mathcal{K}$)
23:          $n \leftarrow$ MatchLength(**draft**, **verified_draft**)
24:          **accepted_tokens** $\leftarrow$ **verified_draft**$[1 \ldots n]$
25:          **if** $n = $ len(**draft**) **then**                     $\triangleright$ All drafts accepted
26:             **accepted_tokens** $\leftarrow$ **accepted_tokens** + **reused_suffix**
27:          **end if**
28:          **prefix** $\leftarrow$ **prefix** + **accepted_tokens**
29:                           $\triangleright$ **Phase 3: Learning and Cache Update**
30:          $r \leftarrow$ ComputeReward(len(**accepted_tokens**), len(**draft**))
31:          Update $\mathcal{Q}$ table using states, actions, and reward $r$.
32:          Update token cache $\mathcal{K}$ with verified sequences.
33:      **end while**
34:      **return prefix**
35: **end function**

---

### C.1  EX-POST TOKEN REUSE AT DRAFT AND TARGET MODELS

The specified attention masks in Figure 6 support branching token sequences by allowing multiple continuations to share the same prefix and draft while maintaining isolated attention within each branch. In both level I (see Figure6a) and level II (see Figure 6b) cache reuse process, the attention mask ensures that all branches can attend to the common prefix and draft tokens—enabling efficient reuse of computations—while preventing any branch from accessing tokens generated by other branches. This isolation is achieved by partitioning the attention mask into distinct regions (e.g., cache 2-1, 2-2) where each continuation only attends to its own tokens and the shared context, thus enabling parallel exploration of different output paths from a common history without interference.

The ex-post token reuse mechanism is attached to both the draft and target models to maximize token utilization. It operates in two distinct modes, corresponding to the drafting and verification stages.

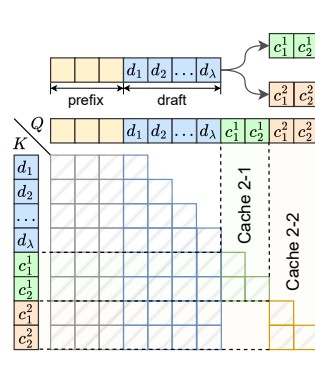 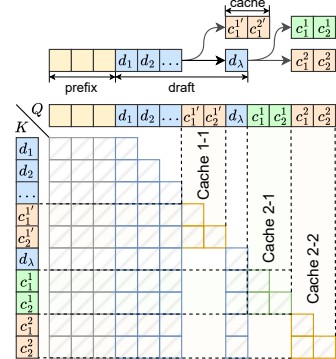

(a) Attention mask for drafting      (b) Attention mask for verifying

Figure 6: Specified attention masks

---

**Algorithm 2** Ex-Post Token Reuse at Draft Model

---
1: **function** REUSEATDRAFT(**prefix**, **draft**, $\mathcal{K}$, $\mathcal{M}_d$)
2:      Let $\tilde{d}_t$ be the last token of **prefix** + **draft**.
3:      Retrieve $W$ candidate phrases $\{\tilde{c}^j_{1\ldots\beta}\}^W_{j=1}$ from $\mathcal{K}$ where $\tilde{c}^j_0 = \tilde{d}_t$.
4:      Construct input sequence: **prefix** + **draft** + $\tilde{c}^1_{1\ldots\beta} + \cdots + \tilde{c}^W_{1\ldots\beta}$.
5:                                              ▷ **Parallel Verification with Draft Model**
6:      Verify sequence with $\mathcal{M}_d$ using attention mask (Figure 6a).
7:      Get verified phrases $\{c^j_{1\ldots\beta}\}^W_{j=1}$ and next token $\tilde{d}_{t+1}$.
8:                                                           ▷ **Find Maximal Reuse**
9:      **for** $j \leftarrow 1$ to $W$ **do**
10:          $A_j \leftarrow$ MatchLength($\tilde{c}^j_{1\ldots\beta}, c^j_{1\ldots\beta}$)
11:      **end for**
12:      $j^* \leftarrow \arg\max_j A_j$, and $\hat{A} \leftarrow A_{j^*}$.
13:                                                                ▷ **Append to Draft**
14:      **if** $\hat{A} > 1$ **then**
15:          Append $c^{j^*}_{1\ldots\hat{A}}$ to **draft**.
16:      **else**
17:          Append $\tilde{d}_{t+1}$ to **draft**.
18:      **end if**
19:      Update $\mathcal{K}$ with verified phrases $\{c^j_{1\ldots\beta}\}$.
20:      **return draft**
21: **end function**

---

**Reuse at Draft Model (Algorithm 2):** This process aims to extend the current draft sequence using cached tokens, reducing the number of auto-regressive steps required from the draft model.

1. **Cache Retrieval:** Given the current draft, retrieve $W$ candidate phrases from the token cache $\mathcal{K}$ that start with the last token of the draft.

2. **Parallel Verification:** Concatenate the current draft with all candidate phrases and process them in a single forward pass of the draft model $\mathcal{M}_d$. A specialized attention mask (Figure 6a) ensures each candidate is verified independently against the shared prefix. This pass yields both verified versions of the cached phrases and the next standard auto-regressive token $\tilde{d}_{t+1}$.

3. **Optimal Phrase Selection:** Compare each original cached phrase $\tilde{c}^j$ with its verified version $c^j$ to find the one with the longest matching prefix, $\hat{A}$.

4. **Draft Extension:** If the best match $\hat{A}$ is longer than one token, append the verified matching sequence to the draft. Otherwise, append the single next token $\tilde{d}_{t+1}$. This ensures at least one token is added per forward pass.

5. **Cache Update:** The newly verified phrases are stored back into the cache, refreshing it with more accurate sequences.

**Reuse at Target Model (Algorithm 3):** This process occurs during the main verification step to append additional tokens if the entire draft is accepted, further increasing the number of tokens generated per target model call.

---

**Algorithm 3** Ex-Post Token Reuse at Target Model

---

1: **function** REUSEATTARGET(**prefix**, **draft**, $\mathcal{K}$, $\mathcal{M}_t$)
2:     Let anchor tokens be the last $m$ tokens of **draft**: $\{\tilde{d}_{\lambda-m+1}, \ldots, \tilde{d}_\lambda\}$.
3:     For each anchor $\tilde{d}_k$, retrieve $W$ candidate phrases $\{\tilde{c}_{1\cdots\beta}^{k,j}\}_{j=1}^W$ from $\mathcal{K}$.
4:     Construct input by inserting candidate phrases after their respective anchors in the draft.
5:                                                    ▷ **Parallel Verification with Target Model**
6:     Verify the full sequence with $\mathcal{M}_t$ using attention mask (Figure 6b).
7:     Get verified draft $d_{1\cdots\lambda}$ and verified phrases $\{c_{1\cdots\beta}^{k,j}\}$.
8:                                                      ▷ **Find Maximal Acceptance**
9:     $n \leftarrow \text{MatchLength}(\tilde{d}_{1\cdots\lambda}, d_{1\cdots\lambda})$.
10:    **accepted_tokens** $\leftarrow d_{1\cdots n}$.
11:    **if** $n < \lambda - m$ **then**                                     ▷ Early rejection
12:        **return accepted_tokens**
13:    **else if** $\lambda - m \le n < \lambda$ **then**                    ▷ Intermediate rejection
14:        Find best reuse phrase $c^{\text{best}}$ at anchor $\tilde{d}_n$.
15:        Append $c^{\text{best}}$ to **accepted_tokens**.
16:    **else**                                    ▷ $n = \lambda$, full draft accepted
17:        Find best reuse phrase $c^{\text{best}}$ at anchor $\tilde{d}_\lambda$.
18:        Append $c^{\text{best}}$ to **accepted_tokens**.
19:    **end if**
20:    Update $\mathcal{K}$ with all verified phrases.
21:    **return accepted_tokens**
22: **end function**

---

1. **Multi-Key Cache Retrieval:** Instead of one, select the last $m$ tokens of the draft as anchors. For each anchor, retrieve multiple candidate phrases from the cache. This multi-anchor approach increases the chances of finding a reusable sequence.

2. **Hierarchical Verification:** Construct a single, long sequence by inserting the retrieved phrases after their corresponding anchors in the draft. This sequence is verified by the target model $\mathcal{M}_t$ in one pass, using a hierarchical attention mask (Figure 6b) that isolates each phrase group.

3. **Conditional Acceptance:** First, determine the number of accepted draft tokens, $n$.

   - If the draft is rejected early ($n < \lambda - m$), no cached tokens are appended.
   - If the draft is partially accepted up to an anchor point ($\lambda - m \le n < \lambda$), the system appends the best-matching verified phrase corresponding to that anchor.
   - If the entire draft is accepted ($n = \lambda$), the system appends the best-matching phrase from the final anchor, maximizing the output length.

4. **Cache Update:** All phrases verified by the target model are used to update the cache, ensuring it contains high-quality, target-approved sequences.

This two-level reuse strategy ensures that token efficiency is maximized at both the drafting and verification stages, forming the core of the ex-post utilization approach.

# D EXPERIMENTAL SETUP AND DETAILS

## D.1 DATASET CONFIGURATIONS

In our experiments, we evaluate the effectiveness of our methodon 4 categories of text generation tasks, including code generation, multi-round dialogue and simple question answering, comparing with other baseline methods.

**Code generation task:** We employ HumanEval (Chen et al., 2021), a famous code generation benchmark which is composed of 164 entries, mainly used for testing model compatibility of our method, as there are various models that are instructed for code generation (e.g. CodeLlama).

**Arithmetic reasoning inference:** We employ GSM8K (Cobbe et al., 2021) and as the evaluation benchmark. For GSM8K, we sample the first 100 entries for evaluation.

**Multi-round dialogue task:** We employ MT-bench (Zheng et al., 2023) as the benchmark. This dataset contains different types of multi-round dialogue tasks including writing, roleplay, reasoning, math, coding, extraction, stem and humanities. We use this dataset for test the task compatibility of ours and other baseline methods.

**Question answering:** we employ Alpaca (Taori et al., 2023) as the benchmark, this dataset contains a series of simple questions that can be used for evaluating details performance of models / methods (e.g. token utilization rate) as the difficulty of different tasks in this dataset is similar.

## D.2 MODEL CONFIGURATION

We select several state-of-the-art LLM families to investigate the effectiveness of our proposed method, including CodeLlama (Roziere et al., 2023), Llama 2 (Touvron et al., 2023) and Llama 3.1 (Grattafiori et al., 2024), the configuration of all models are listed in Table 6. Our methoddoes not introduce any additional training, and directly uses these models to evaluate our algorithm.

| Models | Layers | dim | FFN dim | Inference throughput (tokens/s) |
|---|---|---|---|---|
| CodeLlama-7B | 32 | 4096 | 11008 | – |
| CodeLlama-34B | 48 | 8192 | 22016 | 8.98 |
| CodeLlama-70B | 80 | 8192 | 28672 | 4.56 |
| Llama-2-7B | 32 | 4096 | 11008 | – |
| Llama-2-70B | 80 | 8192 | 28672 | 4.54 |
| Llama-3.1-8B | 32 | 4096 | 14336 | – |
| Llama-3.1-70B | 80 | 8192 | 28672 | 4.57 |

Table 6: The model configuration of the evaluated models.

## D.3 DEVICE AND INFERENCE CONFIGURATION

All of our experiments including compatibility study, ablation studies, and case studies are conducted on 4×NVIDIA A6000 48G GPUs. For maximum inference throughput for both draft and target models, we load the model in *bfloat16* precision and use *eval()* method. The draft model is loaded to single A6000 GPU with *auto* load configuration, while the target model is loaded to 4×A6000 GPUs with *auto* load configuration. For inference, we use batch size 1, which is commonly used in other Speculative Decoding works.

## D.4 HYPER-PARAMETERS

The experimental configuration employs consistent hyper-parameters across all comparative methods to ensure equitable benchmarking conditions. We adopt the default parameter settings specified in the original publications of each baseline approach, including Lookahead decoding and Ouroboros. For scenarios where specific task-model combinations lack explicit parameter specifica-

tions in the reference literature, we implement the most widely adopted configurations documented in their respective studies.

### D.4.1 MAXIMUM GENERATION LENGTHS

The maximum generation lengths for different experimental scenarios are systematically configured according to task requirements, as documented in Table 7. We standardize on 256 tokens for general instruction-following tasks (alpaca dataset) and complex dialogue evaluations (MT-bench), while adopting 128 tokens for code completion benchmarks (Humaneval) to align with common practice in programming language generation research.

| Experiment | Dataset | Maximum generation length |
|---|---|---|
| Motivation Experiment | alpaca | 256 |
| Overall test on Humaneval | Humaneval | 128 |
| Overall test on MT-Bench | MT-bench | 256 |
| Overall test on GSM8K | GSM8K | 128 |
| Ablation Study | Humaneval | 128 |
| Draft Length Sensitivity Study | Humaneval | 128 |
| Draft Length Sensitivity Study | MT-bench | 256 |
| Mean Accepted Token Length | Humaneval | 128 |
| Token Utilization | Humaneval | 128 |

Table 7: Maximum generation length for different experiments and datasets.

### D.4.2 HYPER-PARAMETERS FOR SPECULATIVE DECODING

For conventional Speculative Decoding methods that require fixed draft lengths, we maintain the canonical values established in their original implementations. Traditional approaches typically employ static draft lengths (e.g., $\lambda = 12$ for Speculative and Ouroboros), whereas our our method-dynamically adjusts $\lambda$ within the range [0–24] through its adaptive speculation mechanism. The maximum draft length of 24 tokens represents an empirical upper bound determined through preliminary validation experiments.

### D.4.3 HYPER-PARAMETERS AND DETAILS FOR $Q$-LEARNING

For the $Q$-Learning, $Q_{\text{init}}$ is initialized with the score for action **continue** slightly larger than **stop**, hence initially the draft length will met the maximum allowed one. During the inference, for warming up, we treat virtual continue/stop decision at each position as an individual Q-learning trial, hence the $Q$ Table could be quickly learnt the relationship of confidence score and token efficiency. This procedure does not inject any specific prior knowledge: all updates come solely from real-time acceptance feedback. We use widely used HP for $Q$-Learning listed as follows:

| HP | Discount factor $\gamma$ | Learning rate $\alpha$ | Exploration $\varepsilon$ |
|---|---|---|---|
| Value | 0.99 | 0.1 | 0.5 |

Table 8: Hyperparameters used in the RL algorithm

# E  ADDITIONAL EXPERIMENT RESULTS

## E.1  COMPONENT-WISE WALL-CLOCK FOR SINGLE OPERATION

The following table lists the component-wise wall-clock for single component operation, the test is executed on Llama-2-Chat 7/70B with Humaneval, which is the most widely used model pairs across all experiments.

| Component | $M_d$ forward | $M_t$ forward | cache fetch/verify | roll back | controller step |
|---|---|---|---|---|---|
| wall-clock | 21 ms | 152 ms | 0.1 ms | 2 ms | < 0.1 ms |

Table 9: Wall-clock latency breakdown of key components in the speculative decoding pipeline (avg. per step).

## E.2  APPLY ON LATEST MODELS AND SPEC-BENCH DATASET

We evaluate our proposed method with related based lines on Spec-Bench with Llama-3.1-Instruct, results are shown in Table 10. Note that some baselines are implemented on transformer 4.34 that do not support Llama-3.1.

| Method | Dialog | Trans | Sum | QA | Math | RAG |
|---|---|---|---|---|---|---|
| Auto-regressive | 1.00× | 1.00× | 1.00× | 1.00× | 1.00× | 1.00× |
| Speculative (8) | 2.28× | 2.24× | 1.92× | 2.19× | 2.46× | 1.80× |
| Assisted | 1.76× | 2.52× | 1.48× | 2.26× | 2.21× | 2.10× |
| **Ours** | **2.42×** | **3.30×** | **2.83×** | **2.42×** | **3.36×** | **3.38×** |

Table 10: Speedup comparison across multiple benchmarks. All values are relative to the auto-regressive baseline.

## E.3  REAL SERVING SCENARIOS WITH EAGLE-3

we combining our proposed method with Eagle on vLLM, we found that combining EAGLE and token cache, we can achieve at most a 1.5× speedup compared to EAGLE (1.6× to AR) alone and a 2.3× overall speedup compared to auto-regressive (AR) decoding (LLaMA 3.1-Instruct-8B, blazedit in vllm-bench, batch size 4).

## E.4  COMPARING WITH MORE BASELINE

| Method | Humaneval | GSM8K |
|---|---|---|
| Assisted | 2.46× | 2.38× |
| SpecDec++ | 2.23× | 2.26× |
| **Ours** | **3.23×** | **2.87×** |

Table 11: Speedup achieved on Humaneval and GSM8K benchmarks relative to the baseline.

Within the same-family SD scenario that our method targets, SpecDec++ and assisted decoding are the appropriate adaptive-length baselines, and we now include SpecDec++ in our comparisons. SpecDec ++ does not provide trained draft length controller, hence we conduct evaluation on same model pair/dataset/hardware as its experiment for comparison. The comparison is done on Llama-2-chat-7/70B, which all of compared methods used / support.

