# OpenReview forum: "Holistic Token Efficient in Speculative Decoding: Post-use & Pre-cut"
_ICLR.cc/2026/Conference — Submitted to ICLR 2026_

### Official Review · Reviewer_Qyrp · 2025-10-16

**Soundness:** 3
**Presentation:** 2
**Contribution:** 3
**Rating:** 6
**Confidence:** 3

**Summary:**

This paper introduces a strategy to improve the token efficiency of Speculative Decoding. The proposed method consists of
1. Token cache to recycle useful tokens from previous generation steps
2. RL-based controller to adaptively shorten the draft length when the predicted benefit is low.

Experiments show that this approach achieves a 2.52–3.23× speedup over autoregressive decoding and improves token utilization by over 20% compared to vanilla SD methods across various models and tasks.

**Strengths:**

1. Novel Application of Reinforcement Learning for Adaptive Drafting: The paper introduces an RL-based controller with a Q-table to make dynamic "stop" decisions for drafting. While previous adpative drafting methods do not use RL.
2. Complete Framework: The framework combines Post-use (reusing generated tokens) and Pre-cut (preventing wasteful generation) to address token efficiency from two different angles.

**Weaknesses:**

1. Insufficient Differentiation of the Post-Use Mechanism from Prior Work: The paper describes its Post-use token cache with policies for ranking, admission, and eviction (lines 98). However, the high-level description is conceptually very similar to the "phrase candidate pool" used by the Ouroboros baseline.  The paper does not provide sufficient detail on its cache management algorithms to clearly differentiate this component from existing token reuse techniques, weakening the claim of novelty for the Post-use mechanism.
2. Lack adaptive-length speculative sampling baselines.

**Questions:**

1. The hyperparameter of the RL controller is not disclosed.

---

> ### Author Response · Authors · 2025-11-27
>
> We thank the reviewer for the insightful and valuable comments. We respond to each comment as follows and sincerely hope that our rebuttal could properly address your concerns.
>
> > **W1**: Insufficient Differentiation of the Post-Use Mechanism from Prior Work: The paper describes its Post-use token cache with policies for ranking, admission, and eviction (lines 98). However, the high-level description is conceptually very similar to the "phrase candidate pool" used by the Ouroboros baseline. The paper does not provide sufficient detail on its cache management algorithms to clearly differentiate this component from existing token reuse techniques, weakening the claim of novelty for the Post-use mechanism.
>
> **Our token cache pool is specially designed for Speculative Decoding** to manage token cache between draft model calls.
> - Existing token-cache methods such as Lookahead Decoding are primarily designed for _single-pass_ and *single-model* generation, where the token cache is tightly coupled to the most recent tokens. Consequently, the cached states are not robust to the frequent rollbacks and deletion of rejected tokens that are intrinsic to speculative decoding: once some draft tokens are rejected, the cache no longer matches the surviving prefix, the cache need to be re-filled or totally refreshed (i.e. the implementation of Ouroboros, which directly applied Lookahead to SD), so the token cache does not realize its full potential in speed up speculative decoding.
> - In contrast, our token-cache mechanism is explicitly designed for the speculative decoding, is can tracking how cache entries evolve under accept/reject decisions and flexibly manage them., thereby maximizing cache reuse in speculative decoding.
>
> Moreover, our Post-use mechanism is part of a unified global-optimization framework rather than an isolated candidate pool. It is co-designed with the Pre-cut RL controller to maximize global efficiency through a tight feedback loop: dynamic pruning filters out low-confidence drafts before they enter the cache, ensuring that Post-use operates as a high-quality, cross-iteration memory. The same signals used for adaptive drafting—acceptance statistics, draft cost, and reuse potential—also guide cache admission and reranking, coupling the two modules. As a result, our adaptive-λ strategy learns the optimal collaboration pattern between the drafter and target model across diverse model pairs and tasks, without requiring per-task configuration.
>
> > **W2**: Lack adaptive-length speculative sampling baselines.
>
> Our setting focuses on widely available same-family model pairs (e.g., 7B→70B) and does not rely on a separately trained, semi-independent drafter. Many adaptive draft-length methods—including BanditSpec—are developed for a different problem formulation in which the drafter (e.g. Eagle) is independently trained and optimized, often with tree-structured candidate exploration.
>
> Within the same-family SD scenario that our method targets, SpecDec++ and assisted decoding are the appropriate adaptive-length baselines, and we now include SpecDec++ in our comparisons. SpecDec ++ does not provide trained draft length controller, hence we conduct evaluation on same model pair/dataset/hardware as its experiment for comparison. The comparison is done on Llama-2-chat-7/70B, which all of compared methods used / support.
>
> |           | Humaneval     | GSM8K         |
> | --------- | ------------- | ------------- |
> | Assisted  | 2.46 $\times$ | 2.38 $\times$ |
> | SpecDec++ | 2.23 $\times$ | 2.26 $\times$ |
> | Ours      | 3.23 $\times$ | 2.87 $\times$ |
>
> > **Q1**: The hyperparameter of the RL controller is not disclosed.
>
> For RL controller we use widely used default settings listed as follows:
>
> | HP    | Discount factor $\gamma$ | Learning rate $\alpha$ | Exploration $ε$ |
> | ----- | ------------------------ | ---------------------- | --------------- |
> | Value | 0.99                     | 0.1                    | 0.5             |

---

### Official Review · Reviewer_P8gs · 2025-11-01

**Soundness:** 3
**Presentation:** 3
**Contribution:** 2
**Rating:** 4
**Confidence:** 3

**Summary:**

This paper introduces a holistic token-efficient Speculative Decoding (SD) strategy aimed at addressing the high inference latency in Large Language Models (LLMs) by maximizing both the Mean Accept Length (MAT) and draft utilization (Util), which are the primary determinants of SD speedup. This strategy functions as an on-the-fly, training-free SD plug-in that achieves lossless acceleration by combining two complementary mechanisms: Ex-post utilization (Post-use) and Ex-ante reduction (Pre-cut). The Post-use mechanism employs a token cache to recycle and reuse "USEFUL" tokens—those successfully verified by the target model—in subsequent draft and target model forward passes, thereby reducing the number of auto-regressive steps and increasing the yield from already generated drafts.

**Strengths:**

1. The paper introduces a holistic token-efficient SD strategy that combines two complementary mechanisms: Ex-post Utilization (Post-use), which reuses already generated content through token caching, and Ex-ante Reduction (Pre-cut), which adaptively controls draft length to avoid generating unnecessary tokens.
2. The method directly addresses the key problem of token wastage in SD while aiming to maximize token utilization and maintain high mean accepted length.
3.  Extensive experiments verify the effectiveness and generality of the proposed method.

**Weaknesses:**

1. The Pre-Cut strategy uses the product of draft token confidences as a predictor of expected accepted tokens, but the paper does not explain why this metric is preferable to alternatives (e.g., weighted average). A theoretical rationale or empirical comparison would strengthen the motivation for this choice.
2. In Table 4, the token utilization rate for the CodeLlama 7B → 70B pair is only 6.23%, which is drastically lower than all other results (typically 64–74%). If correct, this requires a dedicated explanation in Section 6.5 to clarify the cause of such extreme inefficiency.
3. The text on lines 54–55 (e.g., “Token Cache (Post-Use)”) appears directly above Figure 1 in a visually awkward position. Moving this text below Figure 1 would improve readability and professionalism.
4. Section 4 should explicitly reference the correct subfigures in Figure 2. The discussion of the re-generated draft token ratio (near line 235) should cite Fig. 2(b), and the confidence product vs. acceptance rate correlation (near line 239) should cite Fig. 2(c).

**Questions:**

1. Adding a high-level architectural overview figure early in the paper would greatly improve clarity and help readers understand the contributions.

---

> ### Author Response · Authors · 2025-11-27
> **Response to reviewer P8gs (Part 1/1)**
>
> We thank the reviewer for the insightful and valuable comments. We respond to each comment as follows and sincerely hope that our rebuttal could properly address your concerns.
>
> > **W1**: The Pre-Cut strategy uses the product of draft token confidences as a predictor of expected accepted tokens, but the paper does not explain why this metric is preferable to alternatives (e.g., weighted average). A theoretical rationale or empirical comparison would strengthen the motivation for this choice.
>
> We examined several alternatives like average confidence and entropy-based measures but the confidence product proved to be the most aligned with the actual situation of speculative decoding. SD failures are dominated by single-token breakdowns: a single low-confidence token causes the latter drafted segment to be rejected. The confidence product is uniquely sensitive to such break points, while remaining simple enough for a training-free controller. In contrast, average confidence smooths out local minima and cannot distinguish sequences that differ only by a single weak token, and entropy-based metrics require full-distribution statistics and additional normalization, making them noisy and computationally impractical for online updates.
>
> > **W2**: In Table 4, the token utilization rate for the CodeLlama 7B → 70B pair is only 6.23%, which is drastically lower than all other results (typically 64–74%). If correct, this requires a dedicated explanation in Section 6.5 to clarify the cause of such extreme inefficiency.
>
> We thank the reviewer for pointing this out. The 6.23% value in Table 4 is a formatting/display issue during typesetting: the intended value is 62.3%, which is consistent with the utilization ranges (64–74%) observed for other model pairs. We will fix the formatting and update.
>
> > **W3**: The text on lines 54–55 (e.g., “Token Cache (Post-Use)”) appears directly above Figure 1 in a visually awkward position. Moving this text below Figure 1 would improve readability and professionalism.
> > **W4**: Section 4 should explicitly reference the correct subfigures in Figure 2. The discussion of the re-generated draft token ratio (near line 235) should cite Fig. 2(b), and the confidence product vs. acceptance rate correlation (near line 239) should cite Fig. 2(c).
> > **Q1**: Adding a high-level architectural overview figure early in the paper would greatly improve clarity and help readers understand the contributions.
>
> We thank the reviewer for the highly valuable and constructive suggestions regarding the manuscript's presentation. We agree that these changes will significantly enhance the paper's professionalism and accessibility and will implement the following improvements in the final version.
>
> - To improve the visual hierarchy and overall readability, we will relocate the introductory text (e.g., "Token Cache (Post-Use)" on lines 54–55) to a position **below Figure 1**, resolving the awkward layout noted by the reviewer.
> - For Section 4, we will revise the text to include **explicit parenthetical references** to the correct subfigures to clarify the evidence supporting our discussion: the analysis regarding the re-generated draft token ratio (near line 235) will explicitly cite **Fig. 2(b)**, and the discussion regarding the confidence product vs. acceptance rate correlation (near line 239) will explicitly cite **Fig. 2(c)**.
> - About the system overview, we appreciate the reviewer’s suggestion. The first page of the paper is primarily dedicated to background, motivation, and a concise statement of contributions, and its space is tightly constrained by the conference template. We therefore placed the architectural overview at the top of the second page, directly beneath the contribution summary, where it can be displayed clearly and without compression. This placement ensures full visibility of the system diagram while preserving the required structure of the first-page introduction.
>
> We have updated the PDF in respond of your suggestions with all updated marked in red.

---

### Official Review · Reviewer_8ZKg · 2025-11-03

**Soundness:** 2
**Presentation:** 3
**Contribution:** 1
**Rating:** 2
**Confidence:** 4

**Summary:**

The paper proposes holistic token-efficient speculative decoding (SD) by utilizing token cache and adaptively choosing draft length for verification using trajectory confidence. The author aim at utilizing already drafted tokens rather than improving the draft models or darfting itself which is another axis in SD. The result shows improved speed-ups compared to the vanila SD methods.

**Strengths:**

* The paper investigates another-axis for improving speculative decoding other than improving acceptance rate or better training recipe for drafter.
* The proposed method is training-free and without extra computations.
* Presentation is clear and ablation are properly studied.

**Weaknesses:**

**Novelty of the method** : The proposed method basically combines two methods for better utilization of the drafted tokens. However, token reusing is already investigated in Ouroboros [1] as mentioned by the authors, difference between [1] and proposed token-caching strategy is not clearly stated and Table 1 shows only marginal improvements of the proposed method compared to Ouroboros even with pre-cut method. For pre-cut method iteself, there are many training-free methods for adaptively choosing draft length ([2], [3]). While confidence-based q-learninig method is interesting, comparison with other methods for adaptive draft-length is laccking and correlation between confidence and trajectory quality is already well-investigated in the literature so its not entirely novel ([4]).

**Lack of experiments** : The algorithm should be tested on more datasets to prove its efficacy. Moreover, proposed method should be tested with the latest SD methods like in EAGLE-3 [5] (Appendix E only provides only theoretical analysis which might be far from the actual improvements). I recommend authors to test the method on other datasets in Spec-Bench [6].

**Real serving scenarios** : I think utilizing token-caching has critical problem in real-world serving scenario where quries are heterogeneous and non-stationary in-nature. Moreover, it would be good if proposed method is experimented on batched inference.

**Questions:**

Questions

* For the experiment, can authors elaborate on why using greedy-decoding only is for fair comparison (ln 362)? (temperature sampling with SD is also lossless)

* In ln 440, how's the Q_init is initialized? what's the prior knowledge here. If it contains some information in warm-up stage, i think it highly varies for diffrent domains.

* Have you investigated other measures than naive product of confidence (as in Eq. 6) for Q-learning?

[1] (Zhao et al.) Ouroboros: Generating longer drafts phrase
by phrase for faster speculative decoding

[2] (Zhang et al.) Draft Model Knows When to Stop: Self-Verification Speculative Decoding for Long-Form Generation

[3] (Agrawal et al.) AdaEDL: Early Draft Stopping for Speculative Decoding of Large Language Models via an Entropy-based Lower Bound on Token Acceptance Probability

[4] (Fu et al.) Deep Think with Confidence

---

> ### Author Response · Authors · 2025-11-27
> **Response to reviewer 8ZKg (Part 1/2)**
>
> We thank the reviewer for the insightful and valuable comments. We respond to each comment as follows and sincerely hope that our rebuttal could properly address your concerns.
>
> > **W1**: **Novelty of the method** : The proposed method basically combines two methods for better utilization of the drafted tokens. However, token reusing is already investigated in Ouroboros [1] as mentioned by the authors, difference between [1] and proposed token-caching strategy is not clearly stated and Table 1 shows only marginal improvements of the proposed method compared to Ouroboros even with pre-cut method. For pre-cut method iteself, there are many training-free methods for adaptively choosing draft length ([2], [3]). While confidence-based q-learninig method is interesting, comparison with other methods for adaptive draft-length is laccking and correlation between confidence and trajectory quality is already well-investigated in the literature so its not entirely novel ([4]).
>
> Overall, **our method is a joint optimization of global token efficiency**, via simultaneously maximizing Mean Acceptance Length (MAT) and utilization rate of draft tokens (Util) we can reducing both draft- and target-model forward calls for generating same length outputs, hence improve end-to-end performance.
> - With Post-use alone the cache fills with low-value that hurt reuse efficiency, while Pre-cut can improve it via keeping far more reliable reusable segments;
> - With Pre-cut alone the controller trims away high quality segments due to single broken token that cut off the draft, but with Post-use that can be maintained for reuse.
>
> More detailed, **our token cache pool is specially designed for Speculative Decoding** to manage token cache between draft model calls.
> - Existing token-cache methods such as Lookahead Decoding [1] are primarily designed for _single-pass_ and *single-model* generation, where the token cache is tightly coupled to the most recent tokens. Consequently, the cached states are not robust to the frequent rollbacks and deletion of rejected tokens that are intrinsic to speculative decoding: once some draft tokens are rejected, the cache no longer matches the surviving prefix, the cache need to be re-filled or totally refreshed (i.e. the implementation of Ouroboros, which directly applied Lookahead to SD), so the token cache does not realize its full potential in speed up speculative decoding.
> - In contrast, our token-cache mechanism is explicitly designed for the speculative decoding, is can tracking how cache entries evolve under accept/reject decisions and flexibly manage them., thereby maximizing cache reuse in speculative decoding.
>
> Further more, our adaptive-$\lambda$ methods operates under a broader information like draft cost and reuse potential (rather than only where acceptance ends), it can continuously learns the optimal collaborative pattern between the drafter and the target model, making it robust and high-performing across heterogeneous model pairs and tasks without per-task configuration.

---

> ### Author Response · Authors · 2025-11-27
> **Response to reviewer 8ZKg (Part 2/2)**
>
> > **W2**: **Lack of experiments** : The algorithm should be tested on more datasets to prove its efficacy. Moreover, proposed method should be tested with the latest SD methods like in EAGLE-3 [5] (Appendix E only provides only theoretical analysis which might be far from the actual improvements). I recommend authors to test the method on other datasets in Spec-Bench [6].
> > **W3**: **Real serving scenarios** : I think utilizing token-caching has critical problem in real-world serving scenario where quries are heterogeneous and non-stationary in-nature. Moreover, it would be good if proposed method is experimented on batched inference.
>
> Thanks for your advice, following is the full result on Spec-Bench with Latest model Llama-3.1-Instruct 7/70B, where our method can achieve better performance (some baselines like Lookahead and Ouroboros only support Llama-2 model.)
>
> | **Method**      | MT            | Trans         | Sum           | QA            | Math          | RAG           |
> | --------------- | ------------- | ------------- | ------------- | ------------- | ------------- | ------------- |
> | Auto-regressive | $1.00 \times$ | $1.00 \times$ | $1.00 \times$ | $1.00 \times$ | $1.00 \times$ | $1.00 \times$ |
> | Speculative (8) | $2.28 \times$ | $2.24 \times$ | $1.92 \times$ | $2.19 \times$ | $2.46 \times$ | $1.80 \times$ |
> | Assisted        | $1.76\times$  | $2.52\times$  | $1.48\times$  | $2.26\times$  | $2.21\times$  | $2.10\times$  |
> | **Ours**        | $2.42 \times$ | $3.30 \times$ | $2.83 \times$ | $2.42 \times$ | $3.36 \times$ | $3.38 \times$ |
>
> To address you concern about the experimental results for token cache combining with Eagle as well as batched inference, we implemented this on vLLM, we found that combining EAGLE and token cache, we can achieve at most a $1.5\times$ speedup compared to EAGLE ($1.6 \times$ to AR) alone and a $2.3\times$ overall speedup compared to auto-regressive (AR) decoding (LLaMA 3.1-Instruct-8B, blazedit in vllm-bench, batch size 4).
>
> We will update those in the final version.
>
> > Q1: For the experiment, can authors elaborate on why using greedy-decoding only is for fair comparison (ln 362)? (temperature sampling with SD is also lossless)
>
> We follow the practice adopted by the majority of prior speculative decoding baselines we compared.
>
> > Q2: In ln 440, how's the $Q_{\text{init}}$ is initialized? what's the prior knowledge here. If it contains some information in warm-up stage, i think it highly varies for different domains.
>
> $Q_{\text{init}}$​ is initialized with the score for action $\textbf{continue}$ slightly larger than $\textbf{stop}$, hence initially the draft length will met the maximum allowed one. During the inference, for warming, we treat virtual “continue/stop’’ decision at each position as an individual Q-learning trial, hence the $Q$ Table could be quickly learnt the relationship of confidence score and token efficiency. This procedure does not inject any specific prior knowledge: all updates come solely from real-time acceptance feedback.
>
> > Q3: Have you investigated other measures than naive product of confidence (as in Eq. 6) for Q-learning?
>
> Yes, we examined several alternatives like average confidence and entropy-based measures but the confidence product proved to be the most aligned with the actual situation of speculative decoding. SD failures are dominated by single-token breakdowns: a single low-confidence token causes the latter drafted segment to be rejected. **The confidence product is uniquely sensitive to such break points, while remaining simple enough for a training-free controller.** In contrast, average confidence smooths out local minima and cannot distinguish sequences that differ only by a single weak token, and entropy-based metrics require full-distribution statistics and additional normalization, making them noisy and computationally impractical for online updates.

---

### Official Review · Reviewer_fr2t · 2025-11-04

**Soundness:** 3
**Presentation:** 3
**Contribution:** 2
**Rating:** 6
**Confidence:** 4

**Summary:**

This core contribution of this paper is a holistic strategy to improve token efficiency in speculative decoding through two complementary mechanisms.

1. Post-use discussed a strategy to use cache that stores drafted tokens that were verified by the target model. Those tokens will be utilized in later forward passes to avoid regenerating recurring segments. This is a clever way as an alternative to Prompt Lookup Decoding where n-gram is provided by the context itself.
2. Pre-cut learns a lightweight RL controller that sends binary decisions (continue/stop) based on recent acceptance statistics.

Overall, I think this is a good paper that empirically unifies adaptive draft length decision and token caching but the weaknesses put it in a borderline position.

**Strengths:**

1. The motivation is clear, which is to maximize MAT and draft-token utilization.
2. The post-use cache design preserves losslessness and offers a concrete path to multi-token gains per forward pass without training a new drafter.
3. The empirical section covers multiple model families and tasks and reports end-to-end tokens/sec with clear speedup ratios, not just MAT.

**Weaknesses:**

1. The novelty relative to prior token-reuse and adaptive drafting work is not significant, but still I appreciate the empirical efforts for proving the compatibility of both speculative decoding designs.
2. Evaluation is relatively less comprehensive. Only HumanEval, MT Bench and GSM8K results are presented. Consider established benchmarks used in the community such as Spec-Bench.
3. Lack of comparison against other adaptive draft length methods, such as SpecDec++ [1] and BanditSpec [2], especially considering the latter one also frames draft length selection in the RL framework.

[1] SpecDec++: Boosting Speculative Decoding via Adaptive Candidate Lengths

[2] BanditSpec: Adaptive Speculative Decoding via Bandit Algorithms

**Questions:**

1. Can you provide component-wise wall-clock (Md forward, Mt forward, cache fetch/verify, rejection recompute, controller step) and memory overheads? This could help reviewers evaluate the source of the gains.

2. Can you report seeded mean & stdev for Tables 1 to 3?

---

> ### Author Response · Authors · 2025-11-27
> **Response to reviewer fr2t (Part 1/2)**
>
> We thank the reviewer for the insightful and valuable comments. We respond to each comment as follows and sincerely hope that our rebuttal could properly address your concerns.
>
> > **W1**: The novelty relative to prior token-reuse and adaptive drafting work is not significant, but still I appreciate the empirical efforts for proving the compatibility of both speculative decoding designs.
>
> Overall, **our method is a joint optimization of global token efficiency**, via simultaneously maximizing Mean Acceptance Length (MAT) and utilization rate of draft tokens (Util) we can reducing both draft- and target-model forward calls for generating same length outputs, hence improve end-to-end performance.
> - With Post-use alone the cache fills with low-value that hurt reuse efficiency, while Pre-cut can improve it via keeping far more reliable reusable segments;
> - With Pre-cut alone the controller trims away high quality segments due to single boken token that cut off the draft, but with Post-use that can be maintained for reuse.
>
> More detailed, **our token cache pool is specially designed for Speculative Decoding** to manage token cache between draft model calls, with robust implementation.
> - Existing token-cache methods such as Lookahead Decoding [1] are primarily designed for _single-pass_ and *single-model* generation, where the token cache is tightly coupled to the most recent tokens. Consequently, the cached states are not robust to the frequent rollbacks and deletion of rejected tokens that are intrinsic to speculative decoding.
> - In contrast, our token-cache mechanism is explicitly designed for the speculative decoding, is can tracking how cache entries evolve under accept/reject decisions and flexibly manage them.
>
> > **W2**: Evaluation is relatively less comprehensive. Only HumanEval, MT Bench and GSM8K results are presented. Consider established benchmarks used in the community such as Spec-Bench.
>
> Thanks for your advice, following is the full result on Spec-Bench with Latest model Llama-3.1-Instruct 7/70B, where our method can achieve better performance (some baselines like Lookahead and Ouroboros only support Llama-2 model.)
>
> | **Method**      | MT            | Trans         | Sum           | QA            | Math          | RAG           |
> | --------------- | ------------- | ------------- | ------------- | ------------- | ------------- | ------------- |
> | Auto-regressive | $1.00 \times$ | $1.00 \times$ | $1.00 \times$ | $1.00 \times$ | $1.00 \times$ | $1.00 \times$ |
> | Speculative (8) | $2.28 \times$ | $2.24 \times$ | $1.92 \times$ | $2.19 \times$ | $2.46 \times$ | $1.80 \times$ |
> | Assisted        | $1.76\times$  | $2.52\times$  | $1.48\times$  | $2.26\times$  | $2.21\times$  | $2.10\times$  |
> | **Ours**        | $2.42 \times$ | $3.30 \times$ | $2.83 \times$ | $2.42 \times$ | $3.36 \times$ | $3.38 \times$ |
>
> > **W3**: Lack of comparison against other adaptive draft length methods, such as SpecDec++ and BanditSpec, especially considering the latter one also frames draft length selection in the RL framework.
>
> Our setting focuses on widely available same-family model pairs (e.g., 7B→70B) and does not rely on a separately trained, semi-independent drafter. Many adaptive draft-length methods—including BanditSpec—are developed for a different problem formulation in which the drafter (e.g. Eagle) is speciall designed and independently trained, often with tree-structured candidate exploration.
>
> Within the same-family SD scenario that our method targets, SpecDec++ and assisted decoding are the appropriate adaptive-length baselines, and we now include SpecDec++ in our comparisons. SpecDec ++ does not provide trained draft length controller, hence we conduct evaluation on same model pair/dataset/hardware as its experiment for comparison. The comparison is done on Llama-2-chat-7/70B, which all of compared methods used / support.
>
> |           | Humaneval     | GSM8K         |
> | --------- | ------------- | ------------- |
> | Assisted  | 2.46 $\times$ | 2.38 $\times$ |
> | SpecDec++ | 2.23 $\times$ | 2.26 $\times$ |
> | Ours      | 3.23 $\times$ | 2.87 $\times$ |

---

> ### Author Response · Authors · 2025-11-27
> **Response to reviewer fr2t (Part 2/2)**
>
> > Q1: Can you provide component-wise wall-clock (Md forward, Mt forward, cache fetch/verify, rejection recompute, controller step) and memory overheads? This could help reviewers evaluate the source of the gains.
>
> The following table lists the component-wise wall-clock for single component operation, the test is executed on Llama-2-Chat 7/70B with Humaneval, which is the most widely used model pairs across all experiments.
>
> | Component  | $M_d$ forward | $M_t$ forward | cache fetch/verify | roll back | controller step |
> | ---------- | ------------- | ------------- | ------------------ | --------- | --------------- |
> | wall-clock | 21 ms         | 152 ms        | 0.1 ms             | 2 ms      | < 0.1 ms        |
>
> As we mentioned before, the size of token cache is limited and we select an light weight but effective controller hence related computing overhead can be negatable. Meanwhile, since the token cache is stored in terms of tokens ids and the controller is a light weight $Q$-Table ($10^3$ numbers) containing accumulated rewards, the memory overhead, compare with vanilla SD, can be neglected.
>
> > Q2: Can you report seeded mean & stdev for Tables 1 to 3?
>
> We have conducted additional runs on SpecBench and report part of (due to the limit of space ) the mean $\pm$ std over 3 randomly initialized seeds for representative model pairs and tasks below (mean / std of toks),
>
> | **Method**      | MT             | Trans          | Sum            | QA             | Math           | RAG            |
> | --------------- | -------------- | -------------- | -------------- | -------------- | -------------- | -------------- |
> | Auto-regressive | 9.18 / 0.0035  | 9.21 / 0.0042  | 7.72 / 0.0032  | 9.29 / 0.0037  | 9.17 / 0.0022  | 7.80 / 0.0040  |
> | Speculative (8) | 20.96 / 0.0047 | 20.63 / 0.0052 | 14.86 / 0.0040 | 20.34 / 0.0055 | 22.52 / 0.0067 | 14.06 / 0.0036 |
> | **Ours**        | 22.17 / 0.34   | 30.39 / 0.42   | 21.81 / 0.32   | 22.44 / 0.56   | 30.83 / 0.43   | 26.35 / 0.28   |

---

### Meta-Review · Area_Chair_CHYz · 2026-01-07

**Summary:**

This paper proposes a holistic token-efficient speculative decoding framework that combines two complementary mechanisms: Post-use, a speculative-decoding–aware token cache that reuses previously verified draft tokens, and Pre-cut, a lightweight, training-free RL controller that adaptively shortens draft length based on trajectory confidence. The goal is to jointly maximize mean accepted length (MAT) and draft token utilization, thereby improving end-to-end decoding throughput without modifying or retraining draft models.

Across reviewers, there is broad agreement that the paper is soundly implemented, clearly motivated, and empirically demonstrates meaningful speedups over vanilla speculative decoding.

Major Concerns:

The major concern is on the novelty and positioning relative to prior work. Several reviewers view the contributions as largely a combination of existing ideas: token reuse (e.g., Ouroboros / Lookahead-style caching) and adaptive draft-length control. While the authors’ rebuttal clarifies that their cache is explicitly designed to handle accept/reject rollbacks intrinsic to speculative decoding—and that the tight coupling between Pre-cut and Post-use yields nontrivial gains—some reviewers remain unconvinced that this distinction is sufficiently clear or fundamental to warrant strong novelty claims.

Another concern is on the experimental evaluation. Reviewers noted limited benchmark coverage and missing comparisons to adaptive draft-length baselines. Concerns about real-world serving scenarios (heterogeneous queries, batching) were partially addressed with preliminary vLLM experiments combining token cache with EAGLE, though these results are not yet fully integrated into the main paper.

Overall, the paper presents a competent and practically useful systems contribution to speculative decoding, with solid empirical validation after rebuttal. The main weaknesses lie in incremental novelty and clarity of differentiation from prior token-reuse and adaptive drafting methods, which some reviewers still judge as modest despite the authors’ clarifications. I agree with most of the reviewers that the paper would benefit from another around of major revision before it can be published.

**Reviewer Concerns:**

see above

**Reviewer Scores:**

Reviewer scores span a wide range from reject to marginally above threshold, but the median sentiment places the paper near the borderline reject region. The rebuttal addressed some of the concerns. However, the paper is still below the acceptance threshold of ICLR.

---

### Decision · Program_Chairs · 2026-01-26

Reject